# OpenReview forum: "Leftover Lunch: Advantage-based Offline Reinforcement Learning for Language Models"
_ICLR.cc/2024/Conference — ICLR 2024 poster_

### Official Review · Reviewer_u5W8 · 2023-10-16

**Soundness:** 2 fair
**Presentation:** 2 fair
**Contribution:** 2 fair
**Rating:** 5
**Confidence:** 4

**Summary:**

The widely used RLHF algorithmic backbone, PPO, can incur additional computational overhead mode collapse due to its online learning nature. Viewing this problem, this paper proposes an offline policy gradient method, Advantage-Leftover Lunch RL (A-LOL), which optimizes LMs towards desired rewards using only static data. Specifically, A-LOL considers the entire output sequence as a single action, and calculates training data advantage before filtering unfavorable instances.
The proposed A-LOL is easy to implement over standard cross entropy loss by adding sequence-level reward-weighting and importance-sampling weights.
In experiments, the proposed method shows competitive results and data efficiency on our different language generation tasks.

**Strengths:**

1. The proposed method is clear and easy to implement, with relatively few assumptions. Therefore, it may have practical merits.
2. The experiments are relatively throughout and the results are promising.
3. Human study helps demonstrating the efficacy of the proposed method.
4. The paper is generally easy to follow.

**Weaknesses:**

1. Unclear method contribution: the proposed method is nearly, if not exactly, a special case of TRPO/PPO method in the bandit setting. Such a special bandit instantiation has been widely considered in classical RLHF works, such as [1,2].

2. Advantage-weighted policy optimization is also a well-studied method in offline RL. For example, Eq. 5 in this paper is very similar to Eq. 4 in AWAC [3], except for the importance weighting that basically comes from TRPO/PPO.

3. The formulation of considering the entire output sequence as a single action step may suffer from exponentially large action space, which may make policy training harder and less stable. See for example [4, 5]. As an aside, recent works have already tried to learn a per-token reward function that incorporates arbitrary human-designed scoring function(s), which may better cope with the large action space in NLG problem, see, e.g., [6].

4. Weighted behavior cloning has been quite extensively used in prior NLP papers, e.g., [6,7,8,9,10,11]. It will make the algorithmic contribution of this paper more clear if the authors can have a paragraph discussing and comparing with such related works, instead of only citing the CRR paper from offline RL.

5. In Table 2, the comparison with PPO may not be fair, because the reward for PPO is a good-or-bad classifier. The offline RL methods, on the other hand, are fitted towards the original responses that themselves show high linguistic diversity, which would implicitly guide the offline RL methods, especially A-LoL, towards generating longer and more diverse sequences. In short, there is no guiding signal for PPO to generate such sequences, while the offline RL methods implicitly have the guidance.

6. There are several well-established exogenous components in the proposed method, such as (1) importance clipping, (2) discarding negative advantage datapoints, (3) prioritized sampling. It is unclear how each of those exogenous components contribute to the overall performance. It is also unclear if the baselines can also benefit from such exogenous components, e.g., (2) and (3). This again muds the algorithmic contribution of the proposed method and make the experiment results less convincing.

[1] Stiennon, Nisan, et al. "Learning to summarize with human feedback." Advances in Neural Information Processing Systems 33 (2020): 3008-3021.

[2] Ouyang, Long, et al. "Training language models to follow instructions with human feedback." Advances in Neural Information Processing Systems 35 (2022): 27730-27744.

[3] Peng, Xue Bin, et al. "Advantage-weighted regression: Simple and scalable off-policy reinforcement learning." arXiv preprint arXiv:1910.00177 (2019).

[4] Guo, Han, et al. "Text Generation with Efficient (Soft) $ Q $-Learning." (2021).

[5] Snell, Charlie, et al. "Offline rl for natural language generation with implicit language q learning." arXiv preprint arXiv:2206.11871 (2022).

[6] Yang, Shentao, et al. "Preference-grounded Token-level Guidance for Language Model Fine-tuning." arXiv preprint arXiv:2306.00398 (2023).

[7] Govardana Sachithanandam Ramachandran, Kazuma Hashimoto, and Caiming Xiong. 2022. [CASPI] Causal-aware Safe Policy Improvement for Task-oriented Dialogue. In Proceedings of the 60th Annual Meeting of the Association for Computational Linguistics (Volume 1: Long Papers), pages 92–102, Dublin, Ireland. Association for Computational Linguistics.

[8] Feng, Y., Yang, S., Zhang, S., Zhang, J., Xiong, C., Zhou, M., & Wang, H. (2023). Fantastic Rewards and How to Tame Them: A Case Study on Reward Learning for Task-oriented Dialogue Systems. arXiv preprint arXiv:2302.10342.

[9] Norouzi, Mohammad, et al. "Reward augmented maximum likelihood for neural structured prediction." Advances In Neural Information Processing Systems 29 (2016).

[10] Sayan Ghosh, Zheng Qi, Snigdha Chaturvedi, and Shashank Srivastava. How helpful is inverse reinforcement learning for table-to-text generation? In Proceedings of the 59th Annual Meeting of the Association for Computational Linguistics and the 11th International Joint Conference on Natural Language Processing (Volume 2: Short Papers), pages 71–79, 2021.

[11] Marcin Junczys-Dowmunt, Roman Grundkiewicz, Shubha Guha, and Kenneth Heafield. Approaching neural grammatical error correction as a low-resource machine translation task. arXiv preprint arXiv:1804.05940, 2018.

**Questions:**

1. Is the proposed method an offline RL method or (online) off-policy RL method? I am a bit confused by some citations, e.g., "Degris et al., 2012" in the second paragraph of Section 1, which is titled as "Off-Policy Actor-Critic".

2. How does the learned $\pi_\theta$ significantly improve over  $\pi_{ref}$? With the clipping technique in PPO, the learned  $\pi_\theta$ will be constrained within a "$\epsilon$-neighbourhood" around $\pi_{ref}$, which limits the room for possible improvement.
Note that in PPO $\pi_{ref}$ is constantly changing and hence allows the continuous improvement of $\pi_\theta$, but in this paper $\pi_{ref}$ is never updated during training (Appendix A).

3. From Line 2 in Algo. 1, it's unclear how do the authors train the value function $V(x)$. Do the authors sample multiple $y'$ from $\pi_{ref}$ for *each* $x$? If yes, with this multi-sequence sampling, how would the proposed method save compute compared to standard PPO-style LM training? If no, then Line 2 in Algo. 1 will only regress to the reward $R$ of $y'$, which is a crude and high variance estimate of the state value $V(x)$.

4. Will "discarding the data points with negative advantages" worsen data scarcity and harm the quality of the LM generations? For example, even though those data points may be less advantageous with regard to the given reward, they may still be helpful for generating fluent text sequences.

5. Maybe Line 4 in Algo. 1 is "... not converge", instead of "... converges"?

6. How would you justify the approximate importance weight in A-LOL sequence? How is it different from the standard per-step importance sampling in RL? Given that it is the best performing method, it would be important if one can justify it.

7. Could you explain the term $\frac{\ln \pi_\theta}{\ln \pi_{ref}}$ in Eq. 6? Would it be better and easier to optimize if we use the log importance weight $\ln \frac{ \pi_\theta}{ \pi_{ref}} = \ln \pi_\theta - \ln \pi_{ref}$?

8. Will the offline RL methods converge if you only allocate one epoch of training steps?

9. In Fig. 2, is NLL trained on the same number of steps and plotted on the same step counts as other methods that are trained on top of the NLL-based reference policy?

10. How is the training time of value function compared to the training time of policy $\pi_\theta$? Is it fair compared to NLL?

11. Why are preferences inherently unidimensional and cannot be extended to tasks where more than one aspect is important? Why couldn't humans make judgement based on multiple axes?

12. Would the success of A-LoL sequence contradict the single-action assumption?

---

> ### Author Response · Authors · 2023-11-21
>
> We thank the reviewer for their detailed feedback. To clarify, our primary goal is to effectively train language models by exclusively using existing language data and sequence-level language classifiers. Thus, we compare *offline* RL [22] methods that exclusively learn from a static dataset, to reduce training costs and improve efficiency.
>
> __Response to the weaknesses:__
> 1. *"a special case of TRPO/PPO in the bandit setting"*
> We believe this comparison is an oversimplification. PPO/TRPO are online RL that assume the training trajectories are generated by constant interaction with the environment. Our method is offline and assumes access to only static data. Accordingly, value estimate, advantage, and the training process differs significantly from the online methods.
> Thank you for references to PPO in single action bandit setting; we will update Table 1 and the intro. Most modern implementations of PPO for LMs [12,13,14] use per-token action assumption with trajectory level sparse reward (only the last token gets reward). This makes the value estimation of intermediate tokens difficult. Instead of on-policy updates, we use a sequence-level value estimate of reference policy that we cheaply train from its validation performance. To our knowledge, such an easy-to-compute sequence-level value estimate has not been used before for offline RL training of LMs. We also show a combination of algorithmic modifications to improve training efficiency and performance (point 6).
> 2. *"Eq. 5 is very similar to Eq. 4 in AWAC [3],"*
> While it is true that Advantage-weighted Regression [3] is a well-studied method in offline RL, there are other differences from A-LoL. AWR maintains two identical models for value estimate and policy and it trains both models on samples from D. Here, D can collect additional off-policy trajectories. In contrast, A-LoL estimates sequence-level value on top of a frozen LM’s validation performance only once before target policy training. This allows A-LoL to discover a subset of positive advantages within the static training data, since negative advantage instances are detrimental to performance (point 6).
> 3. *"suffer from exponentially large action space"*
> Large action space is not an issue for A-LoL due to its use of offline data (that can be human-written). In contrast, both online on-policy and online off-policy methods use LM-generated data with high variance due to the large action space. While there exist a few studies with per-token [6] and phrase-level [15] reward functions, most prominent RLHF implementations use a sequence-level reward [1,2,12,13,14] and thus, we compare with the same setting.
> 4. *“Weighted behavior cloning has been used in prior NLP papers”*
> Thank you for sharing the related works! We use weighted behavior cloning as a reward-based offline RL baseline and will add a paragraph discussing these additional related works.
> 5. *“reward for PPO is a good-or-bad classifier”*
> Common RLHF benchmarks and PPO implementations [12, 13, 14] use sequence-level preference classifiers as reward models. For a fair comparison, we keep the same initial reference LM and reuse data, and reward models [21] from PRO’s [20] experimental setup (a preference-based offline RL baseline). In the PPO baseline, we use nucleus sampling (top-p=0.95) and also include entropy in the loss term to encourage diversity.
> All offline RL baselines use the same data, but vary in diversity. In particular, DPO [18] loses diversity, whereas all reward-based baselines and A-LoL methods show better diversity.
> 6. *"exogenous components, such as (1) importance clipping, (2) discarding negative advantage datapoints, (3) prioritized sampling.”*
> Thank you for pointing this out. We compare the impact (1) and (3) in the ablations in Appendix B.3 (Figure 6). We find (1) importance clipping is crucial for A-LoL; it degenerates to NaN losses without it. We find (3) priority sampling is notably better than random sampling. We added a new ablation for (2) and found that negative advantage is detrimental.
> For a fair comparison with reward-based baselines, we employed reward-based priority sampling (page 4), but do not discard the negative advantage data as they do not have an internal value estimate like A-LoL.

---

> > ### Author Response · Authors · 2023-11-21
> >
> > __Answers to the Questions:__
> > 1. *"offline RL method or (online) off-policy RL method?”*
> > Our method is offline RL [22]; it only uses the training instances from a static dataset. We convert off-policy equations [23] to offline by assuming single-action and MLE finetuned reference policy on the static dataset.
> > 2. *“How does the learned $\pi_\theta$ significantly improve over $\pi_{ref}$?”*
> > A larger importance weight clipping ($\epsilon$=0.9) was more beneficial in our experiments (Appendix B.3 Figure 6) that allowed the $\pi_\theta$ to drift further away from low-performing $\pi_{ref}$ LM.
> > 3. *"unclear how to train the value function V(x)?”*
> > We sample one output per input in the validation, obtain its reward $R$, and train the value estimate $V(x)$ on a frozen supervised LM with the mean squared error loss. In the HHA task, the $V(x)$ is trained for 10 epochs on the 280 validation instances. We will add these details in the implementation section.
> > 3. *“Will "discarding the data points with negative advantages" worsen data scarcity”*
> > No, it won’t. It is quite common to have noisy, incoherent, or toxic language in datasets. Filtering low-quality data is known to reduce biases and improve output quality in LMs [16, 17]. Our analysis shows negative advantage often corresponds to unfavorable data points (Appendix B.4, Table 12) and discarding them is overall helpful.
> > 4. *“Typo in Algo. 1 ”*
> > Thank you for identifying the typo. We will update the text.
> > 5. *“justify the approximate importance weight in A-LOL sequence”*
> > The approximate importance weight in A-LoL seq. is similar to per-step importance weight in token-level action PPO. However, in PPO, these would also be reweighted with per-token advantages, which can introduce noise if the intermediate value estimates are inaccurate. In A-LoL sequence, we apply a flat instance-level advantage with per-token importance weights. We will update the text to show relevance to token-level RL methods.
> > 6. *Typo in Eq 6*
> > Thank you for pointing this out. We indeed optimize the log of importance weight in our implementation and will update the text.
> > 7. *“Will the offline RL methods converge if you only allocate one epoch of training steps?”*
> > Yes. In the HHA task, we used 7B parameter LMs which potentially don’t need many high-quality instances to converge. We didn’t notice any benefit of running more epochs for DPO or A-LoL in the HHA task. In the smaller LM (355M parameter) Reddit experiments (Section 5), the offline RL methods continued to improve up to 3 epochs.
> > 8. *“In Fig. 2, is NLL trained on the same number of steps as other methods?”*
> > Yes. In the HHA experiment, all offline RL methods, including NLL, start from the same reference LM obtained externally timdettmers/qlora-hh-rlhf-7b. This initial policy was trained for one epoch on the HHA dataset in a previous work [19]. The “+NLL” in our experiment is the second epoch which quickly converges in ~20% steps (Figure 2). Other Offline RL baselines are also trained for same number of steps (except for A-LoL which uses 33% less data points).
> > 9. *"training time of value function compared to the training time of policy $\pi_\theta$?”*
> > Training times of A-LoL variants (Algo 1) finishes a few hours faster than an epoch of NLL (144K instances ~1 day). The value estimation only takes 10 minutes to train using the validation set. Computing the advantages of the training set takes ~4 hours for 144K instances. However, it helps remove the negative advantage instances and reduces the target policy training time (only 97K positive advantage instances used).
> > 10. *“Why are preferences inherently unidimensional?”*
> > Judging on multiple aspects when the attributes don’t have exact alignment can lead to misaligned preferences and makes the label collection more challenging. For example, given a pair of responses the more helpful output can be considered less polite or vice-versa.
> > 11. *“Would the success of A-LoL sequence contradict the single-action assumption?”*
> > This is not true. In the HHA task, humans rate A-LoL model’s responses (with scalar importance weight) more helpful than the A-LoL sequence (per-token importance weight). Their reward performances do not vary significantly across all 4 experiments. The main benefit comes from our advantage term. Hence, even A-LoL reference-free (without importance weight) performs better than reward-based baselines.
> >
> > Additional references:
> > [12] von Werra et al. 2020 - github.com/huggingface/trl
> > [13] Castricato et al. 2023 - github.com/CarperAI/trlx
> > [14] Ramamurthy et al. ICLR (2023) - 2210.01241
> > [15] Wu et al. 2023 - 2306.01693
> > [16] West et al. NAACL (2022) - 2110.07178
> > [17] Swayamdipta et al. EMNLP (2020) - 2009.10795
> > [18] Rafailov et al. 2023 - 2305.18290
> > [19] Dettmers et al. 2023 - 2305.14314
> > [20] Song et al. 2023 - 2306.17492
> > [21] Köpf et al. 2023 - 2304.07327
> > [22] Levine et al. 2020 - 2005.01643
> > [23] Degris et, al ICML (2012) - 1205.4839

---

> > > ### Comment · Reviewer_u5W8 · 2023-11-21
> > > **Response to the authors**
> > >
> > > Dear authors,
> > >
> > > Thank you so much for the detailed response.
> > >
> > > After reading other reviewers' comments and your respective rebuttal. I have some additional questions and comments.
> > >
> > >
> > > 1. > ... such an easy-to-compute sequence-level value estimate has not been used before for offline RL training of LMs
> > >
> > > If I understand correctly, sequence-level value estimate has been used in RL4LMs [14]. In fact, Line 2 of your Algorithm 1 is the same as Line 9 of Algorithm 1 of RL4LMs.
> > >
> > > 2. > ... show a combination of algorithmic modifications to improve training efficiency and performance
> > >
> > > While I appreciate the experiment results, I have to say that those modifications are very standard in previous offline RL and NLP papers, as indecated by the papers listed in my main review. Therefore, I tend to not counting those modifications as the algorithmic contribution of this paper, and would encourage the authors to clearly cite and state the origin of them in the paper.
> > >
> > > 3. I don't quite understand your comparision with AWAC in the rebuttal. My understanding is that the main differences are with problem setting: offline v.s. online off-policy, and stepwise v.s. sequence-wise; rather than algorithmic differences.
> > >
> > > 4. > We sample one output per input in the validation... with the mean squared error loss.
> > >
> > > Thanks for the clarification. Then, as discussed in my main review, how is the estimated $V(\cdot)$ different from $R(\cdot)$?
> > >
> > > 5. > It is quite common to have noisy, incoherent, or toxic language in datasets.
> > >
> > > As far as I understand, low advantage means $R - V < 0$. In other words, it means that the quality of those texts is "below average". It does not mean that those texts are "noisy, incoherent, or toxic language", which is just a sufficient condition but by no means necessary.
> > > My understanding is that the findings in Appendix B.4, Table 12 are tied to the specific datasets that you are working on.
> > >
> > > 6. > In the HHA task, humans rate A-LoL model’s responses (with scalar importance weight) more helpful than the A-LoL sequence (per-token importance weight).
> > >
> > > Maybe I misunderstand something, but it looks to me that in Table 3, the best performing method in terms of Human evaluation is "**A-LOL seq.**", which is also bolded.
> > >
> > >
> > > A small suggestion: for your *Additional references* in the rebuttal, it would be more readable if you could include the name of the paper (like what I did in the review).

---

> ### Author Response · Authors · 2023-11-22
> **Response to the additional questions**
>
> 1. *“sequence-level value estimate has been used in RL4LMs [14]. In fact, Line 2 of your Algorithm 1 is the same as Line 9 of Algorithm 1 of RL4LMs.”*
> It is not correct that RL4LMs use a sequence-level value estimate. RL4LMs makes the per-token action assumption [12,13] and the sequence-level reward arrives at the end of the episode (Section 3.1 in RL4LMs[14], [official code](https://github.com/allenai/RL4LMs/blob/main/rl4lms/envs/text_generation/reward.py#L179-L205)). The per-token reward definition (eq (1) in RL4LMs) also includes a per-token KL penalty. Subsequently, Line 9 in RL4LMs Algorithm 1 is a per-token value estimate from the samples drawn from the masked LM policy and is updated throughout the training. In contrast, our Line 2 in Algorithm 1 estimates a scalar value for the entire input (single-action assumption) on the validation performance of reference LM only once before the training. This reference LM value estimate allows the removal of negative advantage in offline training and enables prioritized sampling.
> 2. *“algorithmic modifications are very standard in previous offline RL and NLP papers”*
> We acknowledge that clipping and reward-based prioritized sampling are modifications suggested by two independent works and are respectively cited as inspiration in Section 2.2 (page 4). However, computing the cheap sequence-level value estimate in an offline setting and removal of negative advantages is novel in our technique. We are able to use advantage-based prioritized sampling because the reference LM advantages stay static. Overall, we tried multiple combinations of these optimizations and developed an offline RL training recipe for LMs that works very well in practice.
> 3. *“comparison with AWAC … the main differences are with problem setting: offline v.s. online off-policy, and stepwise v.s. sequence-wise; rather than algorithmic differences.”*
> There are key algorithmic distinctions in the value estimate of A-LoL and AWAC along with the other mentioned differences in the problem setting. Algorithm 1 lines 5,6 in AWAC [3] uses the samples from off-policy trajectories in $D$ to continually update both the value model and policy model which are two separate networks with identical architectures (Appendix C in AWAC). Instead, A-LoL reuses the bulk of frozen parameters of $\pi_{ref}$ and trains its sequence-level value estimate only once before target policy training and, therefore, is not only more computationally efficient than AWAC but is also able to remove negative advantage data and employ prioritized sampling. Our derivation also naturally yields importance weight in our main equation, which across all our experiments, has shown improvement in the performance and stability.
> 4. *“how is the estimated $V(.)$ different from $R(.)$?”*
> Concretely, we define a value layer (MhA + MLP) on top of the frozen reference LM which only uses the hidden states of any given input $V_{\pi_{ref}}(x)$ and emits a scalar value as output. To train the value estimate, we employ MSE loss on the rewards ($R(x, y’)$) obtained from one reference LM output ($y’ \sim \pi_{ref}(x)$) for all the inputs in the validation set $D_v$ (280 instances in HHA task). Our value function gets a crude estimate of how well the reference LM is expected to behave on any input. In our experiments, the MSE loss of this value estimate approaches 0 but is never equal to 0. Since, the reference LM cannot perfectly reproduce the performance of all the static trajectories in $D_{tr}$ and there is some delta between the rewards of the training set and the value estimate of the reference LM (line 3 of Algorithm 1 in our work).
> 5. *“low advantage means $R - V < 0$. In other words, it means that the quality of those texts is "below average"”*
> Instead of “below average”, we view the negative advantages as instances where reference LM is expected to generate better quality outputs than the ones in the static training set. Intuitively, there should be no benefit of training on these poor quality train trajectories and subsequently removing negative advantage instances yields benefits in all the tasks. We refer to these instances as suboptimal or noisy trajectories. In experiments with known good and bad data splits, A-LoL automatically finds a larger subset of negative advantages in the bad data split (48% of Reddit downvoted comments compared to 36% of Reddit upvoted comments).
> 6. *“in Table 3, the best performing method in terms of Human evaluation is "A-LOL seq."”*
> This is not correct. A-LoL seq. is considered more safe and helpful according to GPT-4, but humans rate A-LoL more helpful than A-LoL seq. (53.5% win vs 49.3% win) and A LoL seq. more safe than A-LoL (63.3% win vs 46.7% win). The discrepancy between GPT-4 and human eval is expected as we saw only 72% alignment in their labels. However, human evaluation should be considered more trustworthy.

---

> > ### Author Response · Authors · 2023-11-22
> >
> > > for your Additional references in the rebuttal, it would be more readable if you could include the name of the paper
> >
> > In order to fit our previous response in 5000 chars we truncated the references. We thank you for the suggestion. Here are all the additional references with titles and links:
> > [12] von Werra et al. 2020 TRL: Transformer Reinforcement Learning https://github.com/huggingface/trl
> > [13] Castricato et al. 2023 trlX: A scalable framework for RLHF https://github.com/CarperAI/trlx
> > [14] Ramamurthy et al. ICLR (2023) Is Reinforcement Learning (Not) for Natural Language Processing: Benchmarks, Baselines, and Building Blocks for Natural Language Policy Optimization https://arxiv.org/abs/2210.01241
> > [15] Wu et al. 2023 Fine-Grained Human Feedback Gives Better Rewards for Language Model Training https://arxiv.org/abs/2306.01693
> > [16] West et al. NAACL (2022) Symbolic Knowledge Distillation: from General Language Models to Commonsense Models https://arxiv.org/abs/2110.07178
> > [17] Swayamdipta et al. EMNLP (2020) Dataset Cartography: Mapping and Diagnosing Datasets with Training Dynamics https://arxiv.org/abs/2009.10795
> > [18] Rafailov et al. 2023 Direct Preference Optimization: Your Language Model is Secretly a Reward Model https://arxiv.org/abs/2305.18290
> > [19] Dettmers et al. 2023 QLoRA: Efficient Finetuning of Quantized LLMs https://arxiv.org/abs/2305.14314
> > [20] Song et al. 2023 Preference Ranking Optimization for Human Alignment https://arxiv.org/abs/2306.17492
> > [21] Köpf et al. 2023 OpenAssistant Conversations -- Democratizing Large Language Model Alignment https://arxiv.org/abs/2304.07327
> > [22] Levine et al. 2020 Offline Reinforcement Learning: Tutorial, Review, and Perspectives on Open Problems https://arxiv.org/abs/2005.01643
> > [23] Degris et, al ICML (2012) Off-Policy Actor-Critic https://arxiv.org/abs/1205.4839

---

> > > ### Comment · Reviewer_u5W8 · 2023-11-22
> > > **Response to the authors**
> > >
> > > Dear authors,
> > >
> > > Thank you so much for your reply. Below are my additional comments.
> > >
> > >
> > > 1. > ...removal of negative advantages is novel in our technique
> > >
> > > It seems to me that this technique comes from Eq. (3) of the CRR paper [24], except that you are dealing with the bandit setting and have added several other techniques.
> > >
> > >
> > > 2. > ... we view the negative advantages as instances where reference LM is expected to generate better quality outputs than the ones in the static training set
> > >
> > > It is unclear to me the rationality and development of this view. I think the practical efficacy of this view is very likely tied to the specific datasets and tasks that you are working on, though it does work in this regard.
> > >
> > > 3. > humans rate A-LoL more helpful than A-LoL seq. (53.5% win vs 49.3% win) and **A LoL seq. more safe than A-LoL (63.3% win vs 46.7% win).**
> > >
> > > Thank you for pointing out this interesting comparison.
> > > If I understand correctly, this comparison coincides with the concern in my original review that the success of A-LoL sequence may *contradict the single-action assumption* --- as we can see, A LoL seq. is more safe than A-LoL.
> > >
> > > [24] Wang, Ziyu, et al. "Critic regularized regression." Advances in Neural Information Processing Systems 33 (2020): 7768-7778.
> > >
> > > ***
> > > Nevertheless, I appreciate the authors' trial of multiple combinations of RL-optimization techniques and the development of an offline RL recipe that practically works well for LM training. I will increase my rating to 5.

---

> ### Author Response · Authors · 2023-11-22
> **Response to additional comments**
>
> We thank the reviewer for considering our responses to the previous concerns and improving the score. However, we would like to address the additional comments raised.
>
> 1. *“removal of negative advantage comes from Eq. (3) of the CRR paper [24], except that you are dealing with the bandit setting and have added several other techniques.”*
> The eq (3) in CRR uses a sign operator on *target* policy advantage as follows $\mathbb{1}[A_{\theta}(s,a) > 0]​$. The authors of CRR mention that *"Intuitively, Eq. (3) entails BC on a filtered dataset where the filtering increases the average quality of the actions we learn from”*. This is different from A-LoL, which uses reference policy advantage estimates and also naturally includes an importance weight term. Thus, the sum total of various components and training procedure of A-LoL is very different from CRR. Moreover, CRR is developed for continuous control tasks where reward signals and data assumptions are very different from the language domain. A-LoL is specifically designed for the large action space and sparse reward signal of language tasks and thus, creates a unique offline training recipe for LMs.
> 2. *“I think the practical efficacy of this view is very likely tied to the specific datasets and tasks that you are working on”*
> We didn’t apply any special criteria in our task selection that would make the removal of negative advantages particularly beneficial. In our experiments, we tested 4 different language generation tasks each with a different set of rewards and data sources - HHA uses an unknown 52B LLM generated responses (Sec 4), Reddit uses human-written internet comments (Sec 5), Commonsense reasoning uses GPT-3 generated data (Appendix C.1) and Knowledge-grounded dialog uses crowd-collected conversational data (Appendix C.2). Depending on the quality of the data source, reward distribution, its supervised reference LM, A-LoL finds 10% to 55% of the data as negative advantage.
> 3. *“the success of A-LoL sequence may contradict the single-action assumption”*
> We would like to reiterate that, although A-LoL seq. improves in diversity and safety, it is not the best-performing method in all the metrics (since humans rate A-LoL more helpful than A-LoL seq. in HHA evaluation Table 3). The net benefit of these different variants of importance weight (full-trajectory, per-token, and KL) is minor compared to the benefit brought by the entire suite of A-LoL methods: changing the reward term with reference LM advantage (as evidenced by all 4 task results in Tables 2, 4, 5, and 6).

---

> > ### Comment · Reviewer_u5W8 · 2023-11-23
> > **Response to the authors**
> >
> > Dear authors,
> >
> > Thank you so much for the reply. Below are my comments to some of your misunderstandings.
> >
> > 1. > The eq (3) in CRR uses a sign operator on target policy advantage...
> >
> > As far as I know, $\mathbb{1}[\cdot]$ stands for the indicator function, not the sign operator.
> >
> > 2. > although A-LoL seq. improves in diversity and safety, it is not the best-performing method in all the metrics
> >
> > I did not claim that A-LoL seq. is the best. I only said that it is successful compared to A-LoL, other A-LoL variants, and the baselines. As I keep saying, the per-token nature of "A-LOL sequence", together with its good empirically results relative to A-LoL, may contradict to the single-action assumption, which is a central backbone to your paper.

---

### Official Review · Reviewer_KNWL · 2023-10-21

**Soundness:** 3 good
**Presentation:** 3 good
**Contribution:** 3 good
**Rating:** 6
**Confidence:** 4

**Summary:**

The paper introduces advantage-leftover lunch RL (A-LoL) which is a class of offline policy gradient algorithms. Crucially, different from most past work in RL+NLP literature, the assumption is that the entire LM output is a single action.

The algorithm is essentially MLE loss (standard cross entropy loss for text generation) but multiplied by sequence-level advantage and importance weight. Algorithm 1 contains the pseudo code. A few tricks are used, including clopping importance weight, advantage priority sampling (simply discard examples with negative advantage). A few variants are proposed
- Regular
- “Ref free” variant (using 1 as importance weights)
- "Sequence” variant (see my concern below)
- “KL” variant (replacing importance weights with KL)

Baselines include PPO (on-policy RL), direct preference optimization (DPO; recent non-RL approach), preference ranking optimization (PRO), and GOLD (offline RL / learning from demonstrations). The generation models are based on llama-7b architecture. The reward models are taken from Pythia (trained on Open Assistant; see footnote 8 and surrounding footnotes). The algorithm is tested on four text generation tasks. The main text includes the helpful harmless assistant task (Anthropic’s dataset), and a Reddit response generation task. There’s improvement over PPO and approaching DPO performance on the helpful harmless assistant task. Different tasks used different baselines.

**Strengths:**

I appreciate the direction of exploring objectives that are not on-policy RL, which is the hot topic these days in fine-tuning LLMs.

I’m glad the discussion of comparison with GOLD exists (Section 2.4), because the motivation and derivation are extremely similar to GOLD (except for a few differences like the per-token action vs. single action distinction, the different treatment of importance weights, etc., as described in Section 2.4). I think it’s totally fine even if it’s similar to GOLD – there are design differences, a few tricks are used, and more experiments are done.

Related to the above point: for experiments, I especially appreciate the experiments on the helpful-harmless assistant task.

**Weaknesses:**

Approximations don’t seem justified mathematically. Maybe it’s alright given RL+NLP research has too many approximations in general – I’ll need to see what other reviewers think.
- For the “ref free” variant: Is it justified to use 1 as importance weights, in the reference free variant of the algorithm? I can’t wrap my head around whether that’s an acceptable approximation, or whether that's making the derived Equation (3) or Equation (4) simply incorrect.
- For the “sequence” variant: The approximation of importance rule is a bit strange. See line 6 of the “variants with alternative importance weight” paragraph on page 4. it’s essentially saying a1 * a2 * … * aT * (b1 + b2 + … + bT) = a1 * b1 + a2 * b2 + … + aT * bT. But this seems wrong? Am I understanding this correctly? Perhaps an explanation of why this is a good approximation will be helpful. But at the same time, the empirical results aren’t really impacted much, so I’m conflicted on how much I should treat this approximation seriously.

A major issue: did the authors train PPO methods for more training steps (more than 1.3 times the training steps of offline methods)? If for more training steps, PPO results improve but your results stay stable, then we can’t say PPO is worse.

Phrasing of the main question in paragraph 1 – the main question seems to be the italicized sentence at the end of the first paragraph: “can we perform rewarded learning, similar to PPO, while only using pre-existing data” but the answer is already yes given the literature in the past few years.
- Cringe loss (https://aclanthology.org/2023.acl-long.493.pdf), as well as the older director loss (https://arxiv.org/pdf/2206.07694.pdf) and unlikelihood loss are relevant. The other algorithms the authors cited are also examples where we can learn from rewards while only using pre-existing data. I think the authors’ research question can be more specific & take prior work into account.
- In addition, I’m also confused about what “similar to PPO” means: do the authors mean that PPO is a form of “rewarded learning” or do the authors mean “can we perform rewarded learning such that the results on X benchmark is similar to PPO performance?”

No on-policy RL performance on Reddit generation task. Is PPO helpful here (given that it’s so popular)?

**Questions:**

Can you elaborate what leftover lunch means?

What amount of training trajectories have importance weights that are larger than 1+epsilon or smaller than 1-epsilon (given the bounds on page 4 in “clipping importance weight”)?

For the HHA task, the authors say that they filtered out 20K out of ~160K of training data with responses that abruptly end in a colon. Can the authors explain why this is helpful/necessary, or give an example?

What kind of tasks would this method not make a difference or fail on? Or does this method work on any text generation task?

---

> ### Author Response · Authors · 2023-11-21
>
> We thank the reviewer for their feedback. We are encouraged that they find our discussion about online vs offline methods that use preference or rewards useful and our motivations and technical contributions clear.
>
> __Responses to the Weaknesses:__
> 1. *“For the “ref free” variant: Is it justified to use 1 as importance weights?”*
> Yes. A-LoL reference free is effectively the supervised LM advantage multiplied with negative log-likelihood loss. Equation-wise, it is very similar to the weighted behavior cloning baseline which is the reward multiplied with the log-likelihood. Another way to look at reference-free is to use the clipping parameter $\epsilon = 0$. A-LoL reference free should be viewed as an ablation as we mainly compare with it to investigate the impact of importance weight.
> 2. *“For the “sequence” variant: it’s essentially saying a1 * a2 * … * aT * (b1 + b2 + … + bT) = a1 * b1 + a2 * b2 + … + aT * bT… an explanation of why this is a good approximation will be helpful…”*
> To clarify, the sequence variant importance weight is an approximation and not mathematically equal. Intuitively, it tries to reweight each token’s likelihood with its token importance weight while still using trajectory-level advantage values. Empirically, we find this approximation to improve diversity in all the tasks while also showing better training convergence than other A-LoL variants. We also find quantitative differences in how the sequence variant affects the importance weights (more in answer to question 2 below).
> 3. *“did the authors train PPO methods for more training steps.”*
> In order to compare the training efficiency of each method, we allocated a similar number of gradient updates to each baseline method including PPO. Our primary goal is to get the best possible policy from existing data while reducing computing costs. We ran for 2.6 times the training steps with PPO but didn’t notice any significant improvement.
> 4. *“Phrasing of the main question in paragraph 1 … is already yes given the literature in the past few years…  I’m also confused about what “similar to PPO” means”*
> We thank you for sharing more related work. The main distinction with previous works is that A-LoL doesn’t need model-generated data during target policy training and can use arbitrary classifiers as rewards. Both cringe loss [1] and unlikelihood training [2] require contrastive trajectories from the learner model during its training loop. Director [3] needs to train the classifier alongside the target LM and cannot use arbitrary pre-existing classifiers as rewards. By “similar to PPO”, we mean the ability of PPO to incorporate arbitrary non-differentiable rewards while using pre-existing language data. We will update the text to highlight these distinctions.
> 5. *“No on-policy RL performance on Reddit generation task”*
> We thank you for this suggestion. We conducted additional experiments with PPO on the Reddit task with a sum of five classifiers as a reward and found that the best model achieved total reward of 3.06 (about 0.1 less than our best A-LoL method). However, their corpus diversity turned out to be very bad with most responses reading like “This is my favorite …” or “I don’t know …”. Overall, PPO gets stuck in a local maxima by optimizing three out of five rewards (fluency, safety, and tf-idf scores) and ignored the other two (engagement and upvote probability). We included this discussion with qualitative examples in Appendix C.3 (Tables 7 and 8).

---

> > ### Author Response · Authors · 2023-11-21
> >
> > __Answers to questions:__
> > 1. *“Can you elaborate what leftover lunch means?”*
> > A-LoL has the unique feature of using the supervised finetuned LM’s value estimate to detect the negative advantage training instances. By exclusively training from positive advantage data, our method is robust to noisy instances (for instance, Reddit response generation from downvoted comments - Section 5). To capture this essence of selecting the leftover positive advantage data, we call our method Advantage Leftover Lunch.
> > 2. *“What amount of training trajectories have importance weights that are larger than 1+epsilon or smaller than 1-epsilon?”*
> > Thank you for this interesting question. Additional analysis tracking the changes in importance weight in A-LoL and A-LoL-seq. variants found importance weight in A-LoL (ratio of probs of whole output) is 66% $> 1+\epsilon$, 23% $< 1-\epsilon$ and 11% of the time between the two ($\epsilon = 0.9$). In A-LoL-seq. (with per-token importance weight) we notice importance weight $< 1-\epsilon$ for only 5% of the token and $> 1+\epsilon$ for 19% of the tokens. Thus, it shows that the A-LoL-seq. variant is better able to make use of the importance weight and subsequently shows better output diversity (more details in Appendix B.5).
> > 3. *“For the HHA task, the authors say that they filtered out 20K out of ~160K of training data with responses that abruptly end in a colon.?”*
> > Many instances in HHA had output sequences that abruptly end in the middle of the sentence, for example: “Here are a few options:”. Early experiments including these bad instances led to all LMs generating incomplete answers to some instances, so we filtered them before training.
> > 4. *“What kind of tasks would this method not make a difference or fail on? Or does this method work on any text generation task?”*
> > Thank you for this question. We discussed the limitations of A-LoL in Appendix D. In our preliminary experiments with machine translation task [4], not included in the main text, we found that A-LoL could not improve lexical matching-based metrics when we used multilingual embedding similarity as the reward. Thus, reward functions need to be well-defined and aligned with the evaluation metrics for A-LoL to succeed. A-LoL will also fail to show improvement if the reference policy is already well-tuned on the data.
> >
> > Additional references:
> > [1] Adolphs et. al ACL (2023) - aclanthology.org/2023.acl-long.493/
> > [2] Welleck et. al ICLR (2020) - 1908.04319
> > [3] Arora et. al ACL (2022) - 2206.07694
> > [4] Bojar et. al WMT 2016 - aclanthology.org/W16-2301/

---

> > > ### Comment · Reviewer_KNWL · 2023-11-23
> > > **Responses to "answers to questions"**
> > >
> > > Q1: I see. I wouldn't have got that. Please emphasize why you named your algorithm A-LoL in a conspicuous spot in the paper. Otherwise I think many readers may be confused.
> > >
> > > Q2: Interesting. So the vast majority of the sequences are out of the (1-epsilon, 1+epsilon) region.
> > >
> > > Q4: Interesting. Another quick question: Would the method perform well on tasks that require longer sequences? I'm just curious what the authors' intuition is.
> > >
> > >
> > > I'm keeping my score at the moment.

---

> > ### Comment · Reviewer_KNWL · 2023-11-23
> > **Reviewer responses to "responses to the weaknesses"**
> >
> > Q1 and Q2: Yeah I understand they are approximations. The question is how liberal (& potentially theoretically unjustified?) of an approximation is acceptable. But good to know that the results are good.
> >
> > Q3: "We ran for 2.6 times the training steps with PPO but didn’t notice any significant improvement." -- That's encouraging if true. It's great that the authors' approach outperforms PPO on the chosen tasks.
> > - I think it's very important to include some results in the paper on longer training using the proposed approach vs. PPO.
> >
> > Q5: Thank you for the on-policy RL results on PPO.

---

### Official Review · Reviewer_3aK6 · 2023-10-24

**Soundness:** 3 good
**Presentation:** 4 excellent
**Contribution:** 3 good
**Rating:** 8
**Confidence:** 4

**Summary:**

This paper proposes a better off-policy policy-gradient based algorithm called Advantage Leftover Lunch RL (A-LoL) in the context of Reinforcement Learning from Human Feedback (RLHF). The proposed A-Lol significantly simplifies PPO by using a fixed advantage function and treating each response as a single action. Experiments are carried out mostly in the HHA benchmark and the reddit response generation task, showing the effectiveness and stability of A-lol.

**Strengths:**

The single-action assumption made here is very reasonable in the RLHF setting because the transition kernel in natural language generation is deterministic and trivial, such that the standard RLHF is in effect a contextual bandit problem instead of an RL problem. RL algorithms like PPO are unnecessarily complicated in the standard RLHF setting, so it's nice to see a more stable contextual bandit problem algorithm. The proposed method is different enough from other alternatives.

In the common HHA benchmark, A-LOL beats other recent preference-based offline RL baselines such as DPO and PRO and other common baselines such as weighted behavior cloning and PPO. Experiments on reddit generation task also shows the advantage of the proposed method and its flexibility in terms of optimizing for versatile rewards.

**Weaknesses:**

A-LoL does not seem to perform better than DPO on the HHA benchmark with the common reward function. In particular, A-LoL seems to be less "Helpful" compared to DPO. Is there any explanation on that? The paper seems to be suggesting that there is an issue of reward hacking with the common reward function, is there a concrete example supporting this claim?

In the offline setting where all the data comes from existing offline datasets, the best that we can do seems only to be as good as the best trajectories in the offline datasets. Is it possible to modify A-LoL such that it can continue to improve itself with online data generated by itself and labeled by the reward model?


minor - The single-action assumption might not always hold especially when the dialogue involves multi-step interaction with the users.

**Questions:**

Why is this method called advantage leftover lunch RL?

I'm curious if other contextual bandit algorithms (such as those from https://arxiv.org/abs/1802.04064) can work well too in the RLHF setting?

---

> ### Author Response · Authors · 2023-11-21
>
> We thank the reviewer for their feedback and are encouraged to see that they find our method more straightforward in language settings compared to modern PPO implementations for RLHF. We also appreciate that the reviewer found our method “different enough from other alternatives” and allows “flexibility in terms of optimizing for versatile rewards”.
>
> __Response to weaknesses:__
> 1. *“The paper seems to be suggesting that there is an issue of DPO reward hacking with the common reward function, is there a concrete example supporting this claim?”*
> During our analysis of high-reward responses in the helpful subset of HHA task, we found that responses with list of items were assigned unusually high rewards. For example, even an empty list (“\n- \n- \n- …”) generated by DPO received 0.84 reward. Subsequently, we found DPO to generate a total of 294 responses with lists where A LoL seq. only generated 51 responses. In comparison, the test set only contained 24 responses with lists.
> 2. *“... Is it possible to modify A-LoL such that it can continue to improve itself with online data generated by itself and labeled by the reward model?”*
> Interesting question! In the future, we certainly want to investigate better exploration strategies than a random sampling of current RLHF implementations and eventually improve A-LoL further with high-quality on-policy generations. In this study, we wanted to benchmark and highlight the interesting properties of A-LoL in the offline setting.
> 3. *“The single-action assumption might not always hold especially when the dialogue involves multi-step interaction with the users.”*
> We agree with this issue and will include it in our limitations section in Appendix D.
>
> __Answers to questions:__
> 1. *“Why is this method called advantage leftover lunch RL?”*
> A-LoL has the unique feature of using supervised finetuned LM’s value estimate to detect the negative advantage training instances. By exclusively training from positive advantage data, our method is robust to noisy instances (for instance, Reddit response generation from downvoted comments - Section 5). To capture this essence of selecting the leftover positive advantage data, we call our method Advantage Leftover Lunch.
> 2. *“I'm curious if other contextual bandit algorithms (such as those from https://arxiv.org/abs/1802.04064) can work well too in the RLHF setting?”*
> Thank you for this interesting suggestion. We will explore these methods in our future work.

---

### Official Review · Reviewer_hcnY · 2023-11-07

**Soundness:** 2 fair
**Presentation:** 2 fair
**Contribution:** 3 good
**Rating:** 6
**Confidence:** 3

**Summary:**

Large language models have shown performance improvement due to training them with reinforcement learning (RL) from human feedback. But unlike other paradigms of learning, RL is very data-inefficient. The authors propose to address this issue by introducing a class of algorithms called Advantage-Leftover Lunch RL (A-LOL). The idea of these algorithms is to utilize better the SFT and the underlying data that the RL algorithm uses. In particular, A-LOL algorithms are offline algorithms that do not require any online samples, making their algorithm more sample-efficient than RL.

**Strengths:**

Strengths:
- The motivation of the paper and the technical contributions are clear.
- The authors perform a thorough empirical investigation of their proposed technique across several tasks. The authors conducted various ablation experiments of the proposed idea to show why the algorithm performed well (e.g., R-LOL versus A-LOL).
- The author studies an important question of comparing online and offline policy gradient algorithms.
- The authors also discuss an interesting issue around a subset of RLHF techniques needing human preference data, whereas other techniques do not.

**Weaknesses:**

Weaknesses:
- Given that in language, the transition function is trivial - it is unclear why offline algorithms are more sample-efficient than online algorithms. Offline algorithms assume access to a lot of quality data, while online algorithms can work with small amounts of data and well-designed reward functions.
- The paper relies on the assumption that each token is not an action but instead a sequence is an action, but it is unclear why this assumption matters. Most RLHF techniques optimize policies on sequence-level losses, not token-level losses because the reward functions are defined on a sequence.
- For the A-LOL algorithms to work, there is a set of assumptions that are not explicitly mentioned in the paper that could have a big impact on performance. In particular, if you don't have good data coverage and a good initial policy, then A-LOL will fail, which means that A-LOL and PPO in RLHF have the same assumptions.
- The experiments did not include PPO due to a seed collapsing, but there has been evidence in the literature that this does not happen, which means the authors did not tune this baseline algorithm properly [1].
- The authors claim that their proposed approach is more data-efficient because they filtered out 33% of good responses, but the same procedure can be done for other techniques. However, a similar procedure was not conducted for the baseline algorithms to show that with less data, their proposed approach performs better.

[1] PAIRWISE PROXIMAL POLICY OPTIMIZATION: HARNESSING RELATIVE FEEDBACK FOR LLM ALIGN- MENT  https://arxiv.org/pdf/2310.00212.pdf

**Questions:**

- There seems to be a typo in equation (2) $\mathbb{E}_{\boldsymbol{x} \sim d^{\pi^{ref}}, \boldsymbol{y} \sim \pi\_\theta} [R(\boldsymbol{x}, \boldsymbol{y}, \star)]$ because $y \sim \pi\_\theta$ in the expectation.
- What is the difference between a single action step and trajectory-base RL? Most RLHF algorithms assume we are performing trajectory-based RL and not token-based RL. The reward function is only defined on the trajectory, not the token level.
- Why is PPO, an online policy algorithm, only able to represent action on a token level, but all the offline policy algorithms can represent action on a sequence level?
- Why can you not optimize the PPO  objective with multiple rewards? If so, then how does PPO perform?
- Are you saying that equation (3) and equation (4) are equal?
- In equation (3) are you ignoring the derivative concerning $\pi\_\theta$ in the ratio ($ \nabla_\theta \frac{\pi\_\theta}{\pi\_{ref}}$)?
- Why are the inputs $D_x$ in $D_{tr}$ satisfying this $D_x \subset d^{\pi^{ref}}$? If you are assuming that $\boldsymbol{x}$ is indepdnent of $\pi_{ref}$ then that seems to mean that $D_x = d^{\pi^{ref}}$.
- What is M, h and $A(\pi\_{ref})$ in algorithm 1 line 1?
- Did you run experiments with R-LOL with the advantage of the learner policy instead of the reward?
- Why can't you sum $\pi\_{ref}$ log probabilities and compute the reward for GOLD the baseline?
- Why is A-LOL more data-efficient? You could also filter the data based on pairs with low rewards for the baseline algorithms and train them. But it is hard to understand if your algorithm is more data-efficient without training the baseline algorithm with the same data.
- The average length of ppo is very odd compared to other algorithms. Do you have qualitative outputs to share? Did you include the kl-penalty into the objective?



Missing citations:
- Pairwise Proximal Policy Optimization: Harnessing Relative Feedback For LLM Alignment Wu et al. 2023
- Learning to Generate Better Than Your LLM by Chang et al. 2023
- Calibrating Sequence Likelihood Improves Conditional Language Generation by Zhao et al. 2023

---

> ### Author Response · Authors · 2023-11-21
>
> We thank the reviewer for their feedback. We are encouraged that they find our discussion about online vs offline methods that use preference or rewards useful and our motivations and technical contributions clear.
> __Response to the weaknesses:__
> 1. *“why offline algorithms are more sample-efficient than online algorithms … online algorithms can work with small amounts of data and well-designed reward functions.”*
> This is not always the case. Online RL is only sample efficient if exploration yields high-quality data, and the reward function is well-designed. Exploration with LMs is tricky due to exponentially large output space [2,3] and reward arrives at the end of the output [4]. Instead, offline RL operates on static training data. A-LoL can further filter low-quality instances using negative advantage. In the HHA task we allocate a similar number of gradient updates to all baselines and PPO shows slower growth (Fig 3) compared to offline RL methods (Fig 2).
> 2. *“The paper relies on the assumption that each token is not an action but instead a sequence is an action, but it is unclear why this assumption matters”*
> Most prominent implementations of PPO in RLHF use per-token as action and assign the full sequence reward to the last token [5,6,7]. This makes the value estimate of intermediate actions tricky and also affects the loss values at each token. Thus, concurrent works are investigating entire output as single-action in offline RL [1,8] for LMs. Using the single-action assumption, our method trains the reference LM value-estimate of the entire sequence and freeze it before training target policy, which allows for stable training (Fig 2).
> 3. *“if you don't have good data coverage and a good initial policy, then A-LOL will fail”*
> Thank you for highlighting this point. We agree that a bad initial policy or lack of good data coverage can be detrimental to the performance of A-LoL. We will clarify this in the limitations section (Appendix D).
> 4. *“did not include PPO due to a seed collapsing, but there has been evidence in the literature that this does not happen[1].”*
> We include the average PPO performance for the two out of three seeds (Table 2). We followed the experiment setup from the PRO baseline [10] which used a 1.4B parameter reward model [11] for training and a 6.9B parameter model for evaluation. The reward model used in the provided reference [1] is a 6B parameter and even the reference policy is different from what we used in our experiment, which could explain the difference in PPO’s performance.
> For hyperparameter tuning, we selected one seed and tested with PPO rollout batches in {16, 128} and learning rates in {2e-4, 2e-5, 5e-6, 5e-7} and found the best configuration be batch size=16 and lr=2e-5. Extending these hyperparameters to 3 seeds resulted in one of the seeds collapsing which highlights the instability in on-policy RLHF [1,2,3]. To verify the correctness of our PPO implementation, we also include the PPO checkpoint from an [external source](https://huggingface.co/reciprocate/ppo_hh_pythia-6B), but found our PPO implementation to work better.
> 5. *“authors claim that their proposed approach is more data-efficient because they filtered out 33% … but the same procedure can be done for other techniques”*
> The reward-based and preference-based offline RL baselines do not come with any principled method to filter data on their own. To filter data with rewards, one would need to know the reward landscape of each training dataset and also analyze the low reward instances to pick appropriate thresholds [9], which is out of the scope of this study. A-LoL is unique in its ability to automatically detect the subset of positive advantage in any training set using the sequence-level value estimate of the initial supervised LM. The amount of negative advantage varies for each task, initial LM, and data split, for example, 33% of the response in HHA task (Section 4), 48% of the downvoted, and 36% of upvoted reddit comments (Section 5) and 10%, 39% and 55% of different data splits of knowledge-grounded dialog (Appendix C.2).
>
> __Answers to the Questions:__
> 1. *“There seems to be a typo in equation (2)  because in the $y \sim \pi_{\theta}$ in expectation.”*
> The equation (2) does have $y \sim \pi_{\theta}$ but we rearrange the probabilities to convert it to $y \sim \pi_{ref}$ as follows:
> $\nabla_{\theta} J(\theta) = E_{x \sim d^{\pi_{\text{ref}}}} [\sum_{y \in \mathcal{Y}}R(x, y, \star) \nabla_{\theta}\pi_{\theta}(y|x)]$
> $= E_{x \sim d^{\pi_{\text{ref}}}}[\sum_{y \in \mathcal{Y}} \pi_{ref}(y|x)\frac{\pi_{\theta}(y|x)}{\pi_{ref}(y|x)} R(x, y, \star) \frac{\nabla_{\theta}\pi_{\theta}(y|x)}{\pi_{\theta}(y|x)}]$
> $= E_{x \sim d^{\pi_{\text{ref}}}, y \sim \pi_{\text{ref}}}[R(x, y, \star)\frac{\pi_{\theta}(y|x)}{\pi_{\text{ref}}(y|x)}\nabla_{\theta}\ln \pi_{\theta}(y|x)]$

---

> > ### Author Response · Authors · 2023-11-21
> >
> > 2. *“What is the difference between a single action step and trajectory-base RL?”*
> > We addressed this in response to weakness 2. Single-action step applies a flat advantage to the entire output (A-LoL), whereas per-token RL methods estimate advantages for every token (full reward to last token).
> > 3. *“Why is PPO only able to represent action on a token level?”*
> > This is indeed incorrect as pointed out by reviewer u5W8 that early instantiations of RLHF considered the entire sequence as action [4]. However, most modern standardized RLHF implementations [5,6,7], use per-token action assumption. We will update Table 1 accordingly.
> > 4. *“Why can you not optimize the PPO objective with multiple rewards?”*
> > Thank you for your suggestion. We conducted additional experiments with PPO on the Reddit task with the reward as sum of five classifiers and found that the best model achieved total reward of 3.06 (about 0.1 less than A-LoL). However, their corpus diversity collapsed with most responses of the form “This is my favorite …” or “I don’t know …”. PPO gets stuck in a local max, optimizing three out of five rewards (fluency, safety, tf-idf scores) and ignoring the other two (engagement, upvote probability). We included this discussion with qualitative examples in Appendix C.3 (Tables 7 and 8).
> > 5. *“Are you saying that equation (3) and equation (4) are equal?”*
> > No. Equation (3) is an off-policy update rule that would require off-policy trajectories to train the model. We approximate the expectation to $D_{tr}$ in equation (4) by assuming that the reference policy is supervised finetuned on training data.
> > 6. *“In equation (3) are you ignoring the derivative concerning in the ratio $(\nabla_{\theta} \frac{\pi_{\theta}(\textbf{y}|\textbf{x})}{\pi_{ref}(\textbf{y}|\textbf{x})})$?”*
> > No. As per the derivation in answer 1, importance weight term is outside the gradients.
> > 7. *“Why are the inputs $D_x$ in $D_{tr}$ satisfying this $D_x \subset d^{\pi_{ref}}$?”*
> > Thank you for pointing this out. We will update the text with $D_x = d^{\pi_{ref}}$. However, even with the subset assumption our method will remain unchanged.
> > 8. *“What is $M$, $h$ and $A(\pi_{ref})$ in algorithm 1 line 1?”*
> > $MhA$ represents a multi-head attention module on top of $\pi_{ref}$.
> > 9. “Did you run experiments with R-LOL with the advantage of the learner policy?”
> > We didn’t. The learner policy $\pi_{\theta}$ value estimate continually changes during training and we would need to reestimate it after each update. To preserve offline properties and efficiency, we never estimate the learner policy advantage.
> > 10. *“Why can't you sum $\pi_{ref}$ log probabilities and compute the reward for GOLD?”*
> > GOLD uses per-token log probabilities of $\pi_{ref}$ as the per-token reward. However, in its vanilla form, it cannot incorporate the real-value sequence-level rewards that are defined by one or more classifiers. To ensure a fair comparison with other reward-based offline RL, we retain the per-token importance weight from GOLD and use our flat sequence-level reward.
> > 11. *“Why is A-LOL more data-efficient? You could also filter the data based on pairs with low rewards for the baselines”*
> > As discussed in the response to weakness 5, A-LoL is unique in its ability to inherently find negative advantage data and finds a different subset for each task (ranging from 10% to 55%). The reward-based and preference-based offline RL baselines do not recommend data filtering in their original work.
> > 12. *“The average length of ppo is very odd compared to other algorithms.”*
> > We hypothesize the low response length due to a weaker reward model (1.4B parameter) and initial reference LM which can lead PPO into a local maxima. PPO occasionally generates long responses (Table 11), but mostly generates shorter responses Tables 9,10,12 and 13 in the Appendix.  We included adaptive KL control with initial coefficient of 0.2 (default configuration in PPOTrainer in trl [5]). We also conducted additional experiments with other hyperparameters but noticed that PPO consistently decreased the average length each time (even with #min output tokens=32).
> > 13. Thank you for the additional references, we will include them in the main text.
> >
> > Supporting references:
> > [2] Guo, et al. EMNLP findings 2022 - 2106.07704
> > [3] Snell, et al. ICLR 2023 - 2206.11871
> > [4] Ouyang, Long, et al. NeurIPS (2022) - 2203.02155
> > [5] von Werra et al. 2020 - github.com/huggingface/trl
> > [6] Castricato et al. 2023 - github.com/CarperAI/trlx
> > [7] Ramamurthy et al. ICLR (2023) - 2210.01241
> > [8] Rafailov et al. 2023 - 2305.18290
> > [9] West et al. NAACL (2022) - 2110.07178
> > [10] Song et al. 2023 - 2306.17492
> > [11] Köpf et al. 2023 - 2304.07327

---

### Meta-Review · Area_Chair_wnGQ · 2023-12-06

**Metareview:**

The reviewers like the clarity of the paper, empirical evaluation, and the simplicity of the proposed approach.

However, they also raise concerns about the assumptions made by the algorithm (e.g. data coverage), the authors' correct use of PPO as a baseline, performance compared to other baselines, and one reviewer has significant concerns on novelty (that are not shared by all reviewers).

The authors do a good job in responding to the comments, however not all reviewers seem convinced.

**Justification For Why Not Higher Score:**

There are several concerns that the rebuttal does not seem to convince the reviewers.

**Justification For Why Not Lower Score:**

I believe that there are enough contributions to merit a publication.

---

### Decision · Program_Chairs · 2024-01-16

Accept (poster)